# Bacterial Biofilms and Their Implications in Pathogenesis and Food Safety

**DOI:** 10.3390/foods10092117

**Published:** 2021-09-08

**Authors:** Xingjian Bai, Cindy H. Nakatsu, Arun K. Bhunia

**Affiliations:** 1Molecular Food Microbiology Laboratory, Department of Food Science, Purdue University, West Lafayette, IN 47907, USA; bai16@purdue.edu; 2Department of Agronomy, Purdue University, West Lafayette, IN 47907, USA; cnakatsu@purdue.edu; 3Purdue Institute of Inflammation, Immunology and Infectious Disease, Purdue University, West Lafayette, IN 47907, USA; 4Department of Comparative Pathobiology, Purdue University, West Lafayette, IN 47907, USA

**Keywords:** biofilm, pathogenesis, food safety, *Listeria*, *Salmonella*, *E. coli*, *Pseudomonas*, *Staphylococcus*

## Abstract

Biofilm formation is an integral part of the microbial life cycle in nature. In food processing environments, bacterial transmissions occur primarily through raw or undercooked foods and by cross-contamination during unsanitary food preparation practices. Foodborne pathogens form biofilms as a survival strategy in various unfavorable environments, which also become a frequent source of recurrent contamination and outbreaks of foodborne illness. Instead of focusing on bacterial biofilm formation and their pathogenicity individually, this review discusses on a molecular level how these two physiological processes are connected in several common foodborne pathogens such as *Listeria monocytogenes*, *Staphylococcus aureus*, *Salmonella enterica* and *Escherichia coli*. In addition, biofilm formation by *Pseudomonas aeruginosa* is discussed because it aids the persistence of many foodborne pathogens forming polymicrobial biofilms on food contact surfaces, thus significantly elevating food safety and public health concerns. Furthermore, in-depth analyses of several bacterial molecules with dual functions in biofilm formation and pathogenicity are highlighted.

## 1. Introduction

### Bacterial Biofilms and Food Safety Concerns

Most microbes found in nature exist in biofilms, a well-structured, dynamic, diverse, synergistic and protective microbial community [1,2]. Biofilm formation (Figure 1) on a solid surface is a natural survival strategy of a microbial cell to compete efficiently with others for space and nutrients and to resist any unfavorable environmental conditions. The solid surface may be biotic (meat, produce, oral cavity, intestine, urogenital tract, skin, etc.) or abiotic (floors, walls, drains, equipment, or food-contacting surfaces). Microbes adhere to surfaces by producing an extracellular polymeric substance (EPS) forming a three-dimensional biofilm scaffold. Metaphorically, EPS is the “house” that covers and protects bacteria in biofilms [3]. Although biofilm architecture is solid, protecting bacteria from physical impact, most of the biofilm is still made up of water [4]. EPS makes up the majority of the total dry mass of biofilms. Approximately one-third of the biofilm’s dry weight is bacterial cells, and the remaining weight comes from bacteria-derived molecules, such as polysaccharides, proteins, and DNA, that make up the EPS [5,6]. Biofilms can be comprised of single-species or mixed-species cultures. The composition of bacteria in biofilms is also affected by surface materials, growth conditions, and biofilm maturity [7]. In the food processing environment, biofilm formation threatens food safety since pathogens can be directly transmitted through contact. After transmission, pathogens can also form biofilms on food surfaces. For instance, *Listeria monocytogenes* found on cantaloupe skin caused a multistate outbreak in 2011 [8,9].

Microbial attachment and biofilm formation on solid surfaces provide the advantages of living in a protective scaffold against desiccation, antibiotics, or biocides (sanitizers), ultraviolet radiation, metallic cations, and physical impact from washing and cleaning. For instance, Martins et al. [10] recently showed that urinary tract infections caused by *Staphylococcus saprophyticus* were more resistant to several antibiotics in their biofilm status compared to their planktonic form. Likewise, biofilms of a commonly used model food spoilage bacterium *Lactobacillus plantarum* were more resistant to several biocides, including organic acids, ethanol, and sodium hypochlorite, than in its planktonic state [11]. Bacteria can acquire and/or exchange genetic materials in biofilms. DNA (plasmid) exchange can take place in biofilms through conjugation and transformation [12,13]. In addition, extracellular DNA can retain the electron shuttle molecule that is critical for redox cycling in biofilms [14].

Common biofilm-forming microorganisms vary in different food processing environments, which may include *Listeria monocytogenes*, *Micrococcus* spp., *Staphylococcus* spp., *Clostridium* spp., *Bacillus* spp., *Lactobacillus* spp., *Brochothrix thermosphacta*, *Salmonella enterica, Escherichia coli*, *Serratia* spp., *Campylobacter* spp. and *Pseudomonas* spp. [15,16]. The control of biofilms in a food processing environment faces many challenges. Similar to the use of antibiotic selection for resistant bacteria, disinfectants or sanitizing agents routinely applied in food processing environments can select pathogens that developed resistance to those chemicals [17]. Furthermore, the resistant strains will experience less competition since the diversity of commensal bacteria in those environments is reduced by chemicals.

Pathogen transmission through food results in approximately 2 billion cases and over one million annual deaths globally [18]. In the United States alone, foodborne pathogens are responsible for approximately 48 million illnesses, 128,000 hospitalizations, and 3000 deaths each year, resulting in yearly economic expenses of 78 billion dollars [19]. A core set of 31 bacterial (64%), viral (12%), and parasitic (25%) pathogens have been identified that are responsible for nine million illnesses in the US each year, and the remainder of 39 million illnesses are caused by pathogens or agents whose identities are unknown. A more recent US survey from 2009 to 2015 reported 5760 outbreaks that resulted in 100,939 illnesses, 5699 hospitalizations, and 145 deaths [20]. Further foodborne disease surveillance from 2011 to 2017, in the US, indicated an average of 842 outbreaks every year resulting in approximately 14,237 illnesses [21]. Interestingly, this report concludes that foodborne cases have remained unchanged since 2011 in the US and also in the Republic of Korea [21].

Pathogen transmissions occur primarily through raw uncooked or undercooked foods and by cross-contamination during unsanitary food preparation practices. Pathogens find a harborage site or niche in food production facilities or product surfaces by forming biofilms [22]. These niches serve as a major source of foodborne outbreaks, especially in cafeterias, hospitals, cruise ships, and commercial food processing facilities. For example, the ubiquitous existence of *L. monocytogenes* in nature gives it numerous routes to be introduced in a food processing environment with various fresh produce or raw materials [23,24]. Once *L. monocytogenes* finds a niche in a food processing facility, it can attach to several abiotic surfaces, such as stainless steel, PVC, and polystyrene, and start to form biofilms, which can be resistant to sanitation and may lead to recurrent food contamination [25,26]. Repeated sampling of multiple food processing environments showed that similar *L. monocytogenes* strains can persist for a few months and up to 12 years [27]. The persistence of certain *L. monocytogenes* isolates in the food processing environment may also be due to the same strains that were consistently introduced by raw material, or because of ineffective sanitation practices [28,29].

Therefore, it is essential to understand the physiology and pathogenesis of biofilm-forming or sessile cells and establish effective control measures for their elimination from food production and processing environments, including school cafeterias or other community-based food production facilities. This review compares the relationship between bacterial biofilm formation and their pathogenesis among the four most common foodborne pathogens, *Listeria monocytogenes, Staphylococcus aureus, Escherichia coli,* and *Salmonella enterica* (Table 1). We also discuss pathogenesis of *Pseudomonas aeruginosa* biofilms that is a model used for Gram-negative bacterial biofilm research (Table 1). Most importantly, *Pseudomonas* contributes to polymicrobial biofilm formation with other foodborne pathogens to provide shelter for these pathogens [29,30], and thus its discussion is critical in context with food safety.

## 2. Bacterial Virulence Factors that Contribute to Biofilm Formation and Pathogenesis

Biofilm formation occurs in several stages: (i) attachment, (ii) microcolony formation, (iii) maturation with cellular differentiation, and (iv) detachment or dispersion (Figure 1). In biofilms, microorganisms produce fimbriae, curli, flagella, adhesion proteins, and capsules to firmly attach to a surface [1,6]. Cells grow in close proximity and cell-to-cell communication (quorum sensing, QS) occurs through the production of autoinducers such as N-acyl homoserine lactone (AI-1) or other molecules, which also regulate gene expression for survival, growth, cell density, resistance to antimicrobials, tolerance to desiccation and pathogenesis [38,39]. Understanding the mechanism of quorum sensing in biofilm formation provides an opportunity for the application of appropriate QS inhibitors to control infection and pathogenesis [40,41,42,43,44]. As a microcolony continues to grow, cells accumulate forming a mature biofilm with three-dimensional scaffolding. Loose cells are then sloughed off from a mature biofilm and convert into planktonic cells, which start the life cycle of a biofilm again by attaching to new biotic and/or abiotic surfaces. The cells from biofilms could become a continuous source of food contamination [15]. Virulence factors that are involved in both biofilm formation and pathogenesis are discussed below for *L. monocytogenes, S. aureus, E. coli, S. enterica*, and *P. aeruginosa.*

**Figure 1 foods-10-02117-f001:**
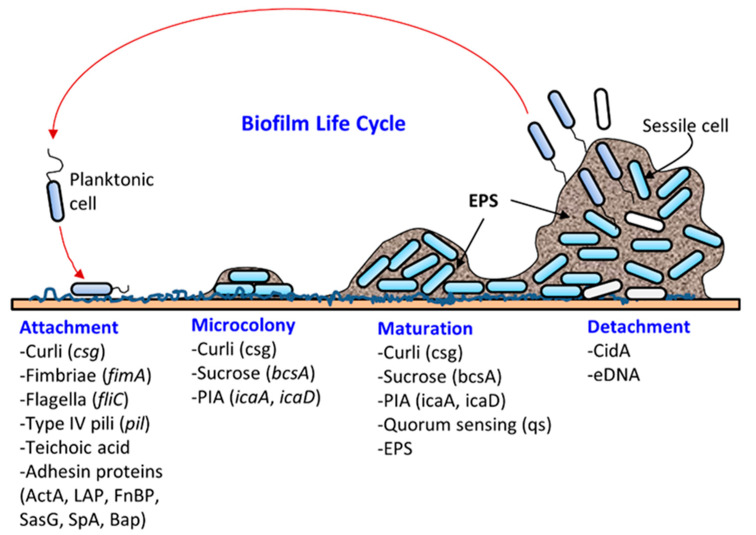
Schematic showing the different stages of biofilm formation (i) attachment, (ii) microcolony formation, (iii) maturation with cellular differentiation, and (iv) detachment or dispersion, and participation of bacterial virulence factors in each step. Abbreviations: ActA, actin polymerization protein; Bap, biofilm-associated protein; bcsA, bacterial cellulose synthesis; CidA, cell death effector protein; csg, curli synthesis gene; EPS, extracellular polymeric substance; eDNA, extracellular DNA; FnBP, fibronectin-binding proteins; icaA, intercellular adhesion; LAP, Listeria adhesion protein; PIA, polysaccharide intercellular adhesin; SasG, *S. aureus* surface protein G; SpA, *S. aureus* protein A. Figure adapted with permission from Ray and Bhunia 2014 [15].

### 2.1. Listeria monocytogenes Cell Surface-Associated Adhesion Molecules Are Involved in Host Cell Interaction and Biofilm Formation

Pathogen interaction with host cells is a prerequisite for initiating the infection process. Several adhesion proteins play a key role in the *L. monocytogenes* infection process and biofilm formation. *L. monocytogenes* is a rod-shaped, Gram-positive, facultative intracellular foodborne pathogen. The gastrointestinal phase of infection is central to it causing localized inflammatory disease (gastroenteritis) or systemic invasive disease [45,46]. As a foodborne pathogen, crossing the intestinal epithelial barrier is a critical step for disease progression. *L. monocytogenes* uses three major invasive pathways: (a) *Listeria* adhesion protein (LAP), (b) Internalin A (InlA), and (c) M-cell-mediated translocation [47] (Figure 2).

(a) In the LAP-dependent pathway, LAP (alcohol acetaldehyde dehydrogenase, AdhE) interacts with the host cell receptor and heat shock protein 60 (Hsp60) [48,49]. It activates NF-kB and myosin light chain kinase to retract epithelial tight junction proteins from the membrane leading to changes in intestinal permeability, and *L. monocytogenes* translocation across the epithelial barrier [50,51].

(b) In the InlA-mediated pathway, surface-expressed InlA anchored to the peptidoglycan interacts with host receptor E-cadherin and promotes transcytosis [52]. Although E-cadherin is one of the cell-to-cell junction proteins located at adherens junctions between epithelial cells, they could be accessible to *L. monocytogenes* in the lumen when cell extrusion or mucus exocytosis creates a transient opening [47,53]. The binding of InlA to E-cadherin (located at the adherence junction) triggers a local cytoskeletal protein rearrangement to trap *L. monocytogenes* into a vacuole followed by cytoplasmic inhabitation for cell-to-cell movement [46]. Though InlA is an important virulence factor, it does not appear to participate in biofilm formation, its expression has been significantly diminished in sessile cells [54].

(c) In the M-cell pathway, *L. monocytogenes* are passively transported by microfold cells (or M cells) in the Peyer’s patch during sampling of luminal antigens [55]. Internalin B, an invasion protein also enhances M-cell-mediated translocation [56] while its direct involvement in intestinal epithelial cell invasion remains controversial [57,58].

After crossing the intestinal barrier, *L. monocytogenes* systemically disseminate to extraintestinal organs, including the spleen, liver, mesenteric lymph node, gall bladder, brain, and placenta in pregnant women [46]. The special strategy of its dissemination is to hide in the cell cytosol and spread to adjacent cells without being exposed to the extracellular environment. Inside the host cell vacuole, *L. monocytogenes* expresses a pore-forming toxin, Listeriolysin O (LLO) (encoded by *hly*), and bacterial phospholipases, PlcA and PlcB to escape from the vacuole. LLO oligomerizes into “arc- or slit-shaped” assemblies in the membrane to lyse the vacuole and facilitate *L. monocytogenes* to escape by the inactivation of phagocyte NOX2 (NADPH oxidase), which produces reactive oxygen species as an antimicrobial strategy [59,60]. Besides, phosphatidylinositol-specific phospholipase C (PI-PLC) and phosphatidylcholine-specific phospholipase (PC-PLC) are also expressed by *L. monocytogenes* in the endocytic vacuole to facilitate escape from the phagosome. The bacterium can also survive in the vacuole for an extended period, causing latent infection [61].

In addition, *L. monocytogenes* also suppresses the cellular proinflammatory response using internalin C (InlC) and moves from cell-to-cell by polymerizing host actin protein (ActA) [62,63]. This triggers host cell cytoskeletal rearrangement and polymerization to propel cell movement towards the host cell membrane [64]. ActA together with PLC helps to avoid autophagy [65,66] by stalling autophagosomal structures [67].

The activation of these virulence genes is regulated under positive control by the master regulator-PrfA (positive regulatory factor). More specifically, the PrfA regulon directly regulates genes in the *Listeria* pathogenicity island-1 (LIPI-1), including LLO, ActA, PlcA, PlcB, Mpl (metalloprotease), and PrfA, and three additional chromosomal loci, which are the *inlAB* operon and *inlC* and *hpt* monocistron. In addition, the expression of as many as 145 other *L. monocytogenes* genes may indirectly be regulated by PrfA [68]. The selective activation of PrfA allows *L. monocytogenes* to convert from an environmental saprotroph to a pathogen by taking on many environmental cues, such as temperature, oxidative stress, carbon sources, and low pH to modulate its virulence [69,70]. PrfA also positively impacts the extracellular environment and mutants lacking *prfA* are defective in surface-adhered biofilm formation [71] possibly due to the impaired regulatory response of key aggregation factors (see below).

#### 2.1.1. Role of Sigma B, PrfA, and ActA on Biofilm Formation and Pathogenesis

Both virulence gene regulators and virulence factors are shown to be involved in biofilm formation in *L. monocytogenes* (Table 2). A stress-response regulator, Sigma B (σB) has been reported to regulate multiple virulence genes and also participates in stress resistance mechanisms when *L. monocytogenes* encounters a harsh environment in the gut and in the environment [72,73]. Sigma B also helps with biofilm formation and provides resistance against sanitizers and disinfectants [74]. Likewise, PrfA also regulates the expression of several virulence factors, including InlA, InlB, ActA, PLC, and LLO, and plays a key role in the physiological transition of *L. monocytogenes* from saprophytic to pathogenic lifestyle [69,70]. Once *L. monocytogenes* infects host cells, PrfA switches to the active status and upregulates the transcription of virulence genes by binding to the promoter regions, referred to as the PrfA box [71]. *L. monocytogenes* with ablated PrfA form significantly less biofilm at 25, 30, or 36 °C; however, the mechanism is not fully elucidated. PrfA regulated ActA is possibly responsible for biofilm formation [75]. Though ActA is involved in the host cell cytoskeletal rearrangement to propel *L. monocytogenes* in the cytosol, it is also reported to be involved in bacterial aggregation and biofilm formation [75]. ActA is also responsible for a sedimentation phenotype, which is essential for bacterial aggregation and biofilm formation. These phenotypes were significantly reduced or absent in strains with deleted *prfA* or *actA* gene which was verified both in vitro cell culture and in vivo animal models (Table 2). Thus, ActA helps *L. monocytogenes* to form aggregates on the surfaces of intestinal epithelial cells and prolongs persistence in mice intestines [75]. This study also reported that the PrfA regulated InlA, InlB, and LLO were neither involved in bacterial sedimentation nor aggregation [75], and thus ruled out their involvement in biofilm formation.

#### 2.1.2. Listeria Teichoic Acid

The teichoic acids are a structural component of Gram-positive bacterial cell wall (peptidoglycan) consisting of alternating phosphate and ribitol (wall teichoic acids, WTA) or glycerol (lipoteichoic acids, LTA) groups, which are replaced with D-alanine and N-acetylglucosamine [76]. LTA or WTA also contributes to biofilm formation in *L. monocytogenes* [77]. Inhibition of N-acetylglucosamine biosynthesis of WTA by tunicamycin (antibiotic) significantly inhibited biofilm formation and reduced *L. monocytogenes* adhesion and invasion to intestinal epithelial cells in vitro. Reduced adhesion and invasion were also attributed to reduced surface localization of InlA, InlB, and LAP due to impaired WTA biosynthesis and destabilization of cell wall architecture [77]. Furthermore, tunicamycin treatment also showed reduced NF-κB activation and inflammatory response due to impaired WTA. This report indicates that WTA is an important virulence factor promoting biofilm formation and pathogenesis in *L. monocytogenes* [77].

#### 2.1.3. Listeria Adhesion Protein

LAP, an adhesion protein [49], has been indirectly attributed to the formation of biofilm (Table 2). Our group has recently shown that recombinant *Lactobacillus casei* expressing LAP from *L. monocytogenes* or *L. innocua* on the bacterial surface showed aggregation and increased biofilm formation on a microtiter plate [51]. In a mouse model, these bioengineered strains also formed thicker biofilms on colonic villi than wild-type *Lactobacillus casei* (Figure 3). Although the function of LAP in the pathogenesis of *L. monocytogenes* has been well documented [48,50], results from recombinant *Lactobacillus casei* highlights the role of LAP in biofilm formation as well.

*L. monocytogenes* form biofilms on food surfaces or food-touching surfaces [9,16]. Therefore, a significant amount of *L. monocytogenes* infection might be caused by those in biofilms. To estimate the virulence potential of *L. monocytogenes* in biofilms, Gilmartin et al. [54] showed that InlA was downregulated in biofilm-isolated *L. monocytogenes* in comparison to its expression in its planktonic counterparts. Recently, our research team [78] showed that not only InlA but LAP and LLO were also downregulated in 48-h-old biofilm-isolated *L. monocytogenes*. Furthermore, those biofilm-isolated bacteria also had attenuated pathogenic phenotypes on cell culture models, including adhesion, invasion, translocation, and cytotoxicity. However, a pathogenicity study using a mouse model demonstrated that although biofilm-isolated *L. monocytogenes* cells had reduced tissue distribution at an early stage of infection (12–24 h), these cells showed upregulated expression of virulence genes in mouse intestinal lumen and eventually caused similar systemic dissemination as planktonic counterparts 48 h post-infection [78]. Even an *L. monocytogenes* murinized InlA (InlA^m^) strain (this strain has a high affinity for mouse E-cadherin) had a similar tissue distribution as the non-murinized (WT) strain in mice at 12–24 h post-infection confirming InlA involvement is limited to early stages of infection. These findings suggest that *L. monocytogenes* cells in biofilms have similar virulence potential as planktonic cells, which highlights the importance of the management of *Listeria* biofilms in food processing environments. In addition, biofilm-isolated cells showed significantly reduced virulence in in vitro cultured mammalian cell models compared to planktonic siblings and provided an inaccurate assessment of biofilm-forming pathogen infectivity. Therefore, animal models are necessary to accurately determine the virulence of biofilm-forming pathogens.

### 2.2. Staphylococcus aureus Surface Proteins Play Important Roles in Biofilm Formation and Pathogenesis

*Staphylococcus aureus* is a Gram-positive opportunistic pathogen and is commonly found in natural waters, soils, animal mucus membranes, and on human skin [79,80]. The anterior nares are a common niche for *S. aureus* [81]. *S. aureus* has a remarkably high salt tolerance, a feature that allows it to be a commensal skin bacterium on approximately 25% of the population, making humans a common vehicle for this pathogen’s transmission to foods.

As a foodborne pathogen, *S. aureus* produce multiple heat-tolerant enterotoxins (at least 24 staphylococcal enterotoxins), toxic shock syndrome toxins, superantigens, exfoliative toxin, and many more virulence factors that can cause food poisoning with symptoms of nausea, vomiting, and stomach cramps [33]. These symptoms usually develop in 30 min to hours after ingestion and are self-limited in most cases. Improperly handled foods are the most common reason for *S. aureus* contamination [82]. In the US, an annual estimate of 241,000 cases of food poisoning is caused by *S. aureus* [19]. Since symptoms in most cases are mild and self-limited, the number of actual infections could be significantly higher than the reported cases. Antibiotics are rarely used for intoxication treatment because *S. aureus* do not usually infect the gastrointestinal tract. However, antibiotic treatment for skin infection has become a great challenge because of biofilm formation and widespread antibiotic resistance.

*S. aureus* contamination of food is a global challenge affecting multiple types of products. Between January 2000 and March 2012, approximately 14 *S. aureus* outbreaks were recorded in Australia, which affected 429 people including 25 hospitalizations and one death [83]. Approximately one-third of the victims were infected after eating at a commercially catered buffet. In Italy, after two individuals suffered from *S. aureus* intoxication, a broad screening of dairy products showed that 102 out of 971 samples were positive for *S. aureus* [84]. Approximately 46% of the isolated *S. aureus* cultures contained at least one enterotoxin coding gene.

*S. aureus* form strong biofilms, which is critical for its persistence not only on food contact surfaces [85] but also on human and animal hosts causing chronic and persistent infections [86,87,88]. *S. aureus* express a large number of surface molecules, which are collectively called microbial surface components recognizing adhesive matrix molecules (MSCRAMMs) that are involved in bacterial adhesion and biofilm formation [88,89]. More specifically, the major biomolecules that are reported to be involved in both biofilm formation and host cell adhesion include fibronectin-binding proteins (FnBP), biofilm-associated protein (Bap), *S. aureus* surface protein G (SasG), *S. aureus* protein A (SpA), cell wall-anchored clumping factor (ClfA), polysaccharide intercellular adhesin (PIA) and teichoic acids (TA), and many have a redundant function [90] (Table 2).

#### 2.2.1. Polysaccharide Intercellular Adhesin (PIA)

PIA was initially discovered in *S. epidermidis* as a critical component responsible for cell-to-cell binding in biofilm formation [91]. Chemically, PIA is a linear β-1,6-linked glycosaminoglycans, and its synthesis is linked to the *ica* (intercellular adhesion) locus, including *icaADB* and *icaC*. Products from *icaA* and *icaD* are responsible for PIA synthesis using UDP-N-acetylglucosamine [92]. Ica proteins of *S. aureus* share 59–78% identity with those proteins from *S. epidermidis* [93]. Deletion of the *ica* locus in *S. aureus* reduced its ability to form multilayer thick biofilms in microtiter plates, suggesting that PIA of *S. aureus* has the same function for cell-to-cell adhesion. The chemical nature of *S. aureus* PIA was identified to be poly-N-succinyl-β-1,6-glucosamine, which is homologous to PIA of *S. epidermidis* [94]. PIA expression is high in oxygen-limiting environments such as seen in cystic fibrosis patients [95]. It is not only critical in biofilm formation but also contributes to pathogenesis. In a mouse model, Rupp et al. [96] reported that intravenous infection by *S. epidermidis* with a deficiency in the *ica* operon did not induce subcutaneous abscess and was possibly eradicated by the immune system. However, a study reported by Francois et al. [97] contradicted these findings. They did not observe any difference in subcutaneous tissue damage in guinea pigs when infected with both wild-type and *ica*-negative *S. aureus* or *S. epidermidis* strains. This suggests that the function of PIA in a pathogen during infection may vary based on different infection methods or models used.

*S. aureus* can also form a biofilm independent of PIA under certain conditions. A two-component system, encoded by *arlRS*, is a repressor of *S. aureus* biofilm formation in Hussain–Hastings–White Modified Medium (HHWm) [98]. Crystal violet stain quantification of biofilm formation in HHWm was double for the *arlRS* mutant strain when compared to the parental strain [98]. Furthermore, deleting the *ica* operon in the *arlRS* mutant strain did not affect *S. aureus* biofilm formation, suggesting that a PIA-independent pathway is responsible for biofilm formation under specific conditions.

#### 2.2.2. *S. aureus* Protein A

Protein A coded by *spa* gene is reported to have a similar function for cell-to-cell adhesin in biofilm development [99]. Mass spectrometry analysis of proteinaceous biofilms components showed protein A as an essential component in biofilm formation by *S. aureus*. Consequently, the *spa* mutant strain exhibited significantly weakened biofilm formation. Furthermore, in an in vitro aggregation experiment, *S. aureus* cells lacking protein A precipitated much slower than a strain expressing protein A, suggesting that protein A may be responsible for *S. aureus* cell-to-cell aggregation [99]. Surprisingly, they also found that protein A was not required to be covalently anchored on the cell surface to function as an adhesin, since the supplemented exogenous protein A could also trigger the aggregation of the *spa* mutant strain [99].

Protein A also serves as an important virulence factor and helps the bacterium to evade the immune system during infection. Protein A is covalently bound to peptidoglycan through an LPXTG motif. It also binds to the Fc region of antibodies, thus forcing antibody binding with the Fab portion facing outward from the cell, reducing the chance of opsonization and phagocytosis [100]. Protein A is also considered a superantigen that binds to the Fab part of the B-cell receptor to induce programmed B cell death (apoptosis) [100].

#### 2.2.3. Biofilm-Associated Protein (Bap)

Another proteinaceous molecule that works as a cell-to-cell adhesin in biofilm formation is the biofilm-associated protein (Bap) (Table 2). Cucarella and colleagues [101], used random transposon insertion mutagenesis and found two mutants with reduced biofilm-forming abilities. Molecular analysis of both mutant strains revealed an insertion in the *bap* gene that encodes Bap, anchored to the cell wall via LPXTG motif. Bap expression is regulated by a global regulator, SarA [102]. An in vivo biofilm formation experiment showed Bap to be responsible for adhesion to inert surfaces. A recent discovery found that low pH and low cation concentration are triggers for the self-assembly of Bap into amyloid aggregates [103]. A mouse foreign body infection experiment revealed that *S. aureus* without Bap caused less persistent infection than wild-type *S. aureus*, suggesting that Bap is a virulence factor responsible for persistent infection [101]. Bap has been a potent target for the dispersion of *S. aureus* biofilms [104].

#### 2.2.4. Fibronectin-Binding Proteins (FnBP)

The fibronectin-binding proteins A and B (FnBPA and FnBPB) are cell-to-surface-anchored (via LPXTG) proteins that bind to host cell fibrinogen and fibronectin and also form biofilm on medical implants and devices [105,106]. FnBPA has been recently shown to be involved in biofilm development in serum and the two-component SaeRS system is required for FnBPA activity [107]. In a mouse model of infection with the absence of FnBPA, *S. aureus* biofilms are more susceptible to immune clearance by macrophages [107]. The FnBPA A domain exerts low-affinity homophilic interaction with neighboring cells to form cell aggregates leading to biofilm formation [108].

#### 2.2.5. *S. aureus* Surface Protein G (SasG)

*S. aureus* surface protein G (SasG) has been shown to participate in biofilm formation and adhesion to epithelial cells [89]. Zinc activates SasG-mediated biofilms by forming a homophilic bond between Zn and SasG protruding from opposite cell surfaces [109,110]. SasG was shown to be involved in adhesion to nasal epithelial cells and its contribution to *S. aureus* biofilm formation happens independently of PIA [89].

#### 2.2.6. Staphylococcal Teichoic Acid

Teichoic acid is a highly charged cell wall polymer that also contributes to biofilm formation in *S. aureus* [76,77]. Ablation of D-alanine synthesis in a *S. aureus* mutant strain affected LTA biosynthesis, and biofilm formation [76]. Likewise, inhibition of N-acetylglucosamine-1-phosphate transferase required for WTA biosynthesis by tunicamycin significantly inhibited biofilm formation by *S. aureus* [77] (Figure 4). Tunicamycin treatment also reduced *S. aureus* adhesion to epithelial cells and reduced NF-κB activation and inflammatory response, suggesting that LTA is an important virulence factor promoting biofilm formation and pathogenesis in this bacterium [77]. More recently, inhibition of LTA by two synthetic inhibitors, HSGN-94 and HSGN-189, showed significant suppression of biofilm formation and synergistic biofilm inhibition when combined with tunicamycin [111].

#### 2.2.7. Miscellaneous Factors

Extracellular DNA (eDNA) is another important component that is involved in biofilm formation independent of PIA in *S. aureus*. The role of eDNA in biofilm formation was first studied in *P. aeruginosa* [112]. The researchers found that the addition of DNase I in the culture of *P. aeruginosa* could effectively reduce biofilm formation without affecting bacterial growth. Interestingly, nicotine (a major component of tobacco smoke) promotes eDNA release and biofilm formation in *S. aureus*; however, it reduces its invasion to alveolar epithelial cells (A549) [113].

CidA (encoded by *cidA* gene), a protein produced by *S. aureus*, is indirectly involved in biofilm formation [114]. CidA is a murein (bacterial peptidoglycan) hydrolase and facilitates the release of eDNA from cells. CidA shows structural similarity to bacteriophage holins involved in phage-induced cell lysis [115]. Mutation of the *cidA* gene affected eDNA release and biofilm formation [116], suggesting that eDNA released by controlled cell lysis is a critical component in *S. aureus* biofilm formation. In addition, the high density of bacteria in biofilms also increases the rate of gene exchange [117]. The frequency of conjugation occurring in biofilms has been reported to be higher than the frequency between planktonic bacteria [118].

### 2.3. Escherichia coli Curli Fimbriae, and Cellulose Are Major Biofilm-Forming Factors

*E. coli* is a Gram-negative, facultatively anaerobic, motile, or nonmotile bacterium and is a natural inhabitant of the intestinal tract of mammals. It has peritrichous flagella, fimbriae or pili, and curli. Some strains form capsules. The majority of *E. coli* strains are commensal and nonpathogenic. However, some are pathogenic and are classified into several pathotypes; enterotoxigenic *E. coli* (ETEC), enteropathogenic *E. coli* (EPEC), enteroinvasive *E. coli* (EIEC), enteroaggregative *E. coli* (EAEC), diffusely adherent *E. coli* (DAEC), adherent invasive *E. coli* (AIEC), and Shiga toxin-producing *E. coli* (STEC) that includes enterohemorrhagic *E. coli* (EHEC) [34]. Epithelial adhesion, colonization, and toxin production (most strains) are essential attributes of pathogens. Among the pathogens, STEC strains can cause lethal foodborne diseases, especially the EHEC strains that colonize the intestinal epithelium and induces attachment-effacement lesions leading to hemorrhagic colitis (HC) with symptoms of bloody diarrhea. Shiga toxin production by these strains can also lead to hemolytic uremic syndrome (HUS), which is often associated with kidney failure and death [119]. Both pathogenic and commensal *E. coli* are reported to form biofilms in the gastrointestinal tract and is orchestrated by regulatory networks, adhesion molecules, and extracellular matrix [120].

*E. coli* outbreaks are not only associated with raw meat or meat products [121] but also with fresh produce [4,122]. For instance, in northern Germany in 2011, organic fenugreek sprouts were contaminated by a novel Shiga-toxin-producing strain of enteroaggregative *E. coli*, which caused 3816 cases of infection including 800 cases of HUS and 53 fatalities [123]. Patients were identified from Switzerland, Poland, Sweden, and even North America, and approximately 800 people had hemolytic uremic syndrome. In a study investigating the survival of *E. coli* O157:H7 on lettuce [124], it was found that approximately 10^5^ CFU/mL waterborne *E. coli* can survive on lettuce for 77 days possibly by forming biofilms on the produce surface. These findings further support the importance of biofilm control in the food processing environment to prevent pathogen contamination.

As one of the pathogens causing the most gastroenteritis cases around the world, *E. coli* is a model bacterium that forms biofilm after well-programmed production of various extracellular molecules [125]. Curli and cellulose are two major components making up the extracellular matrix [126].

#### 2.3.1. *E. coli* Curli

Curli fimbriae are proteinaceous extracellular fibers and are responsible for cell-to-cell and cell-to-surface attachment [127,128]. These amyloid nanofibers are made with repeated protein subunits and encapsulate bacteria in a complex network [129]. The signature phenotype of curli-producing bacteria is red-stained colonies when grown on a medium containing Congo red, which binds to curli amyloid fibers [130].

Curli fimbriae of *E. coli* are composed of self-assembled CsgA nucleated by CsgB that also anchors curli to the bacterial surface [129]. Including the major subunit (CsgA) and nucleation (CsgB), expression of curli fimbriae is achieved by genes in two operons, *csgBAC* and *csgDEFG*, which are responsible for secretion (CsgC and CsgG), transcription regulation (CsgD), and processing (CsgE and CsgF) [131]. The assembly of curli fimbriae requires precipitation of the CsgA monomer with CsgB as the nucleator protein. Transcription of the two operons for curli biosynthesis is regulated by CsgD, OmpR, and sigma factors, σ^70^ and σ^S^ in response to a variety of environmental signals. Under laboratory conditions, the expression of curli is optimal during the stationary phase in an environment with low nutrients and low medium osmolarity at a temperature ≤30 °C [132]. Curli fimbriae are required for initial bacterial adhesion to inert surfaces and the development of biofilms [133]. The CsgA-made nanofibers are highly resistant to heat and detergents [129], which may enhance the persistence of biofilm in the food processing environment. Analysis of sludge samples from several wastewater treatment plants showed that the biovolume of amyloid fimbriae could make up to 10–40% of biofilm volume [134].

In the intestinal tract, curli may not play an essential role in adhesion to the epithelial cells. *E. coli* with a mutation in *csgA* and *csgD* genes showed similar adhesion rates on both HeLa and HT-29 cell lines as the parental strain [135], suggesting that other adhesins may be involved in *E. coli* adhesion to epithelial cells. On the other hand, when CsgD, the curli biosynthesis regulator, was overexpressed, the adhesion of *E. coli* was increased by approximately five-fold, suggesting that curli may still be a redundant adhesion factor. Recently, Elpers and Hensel [136] reported the presence of at least 16 putative fimbrial gene clusters in EHEC (*E. coli* O157:H7, Sakai strain) and a subset of six gene clusters were responsible for adhesion to epithelial cell lines (HeLa, MDCK, and Caco-2) and biofilm formation.

#### 2.3.2. Cellulose

The major exopolysaccharide secreted by *E. coli* in biofilms is cellulose [137]. Cellulose is a (1–4)-β-linked linear glucose chain molecule that can be produced by plants, microorganisms, and some animals and it is the most abundant organic polymer found in nature. After screening 13,000 mutant strains, an unknown substance secreted by *E. coli* was closely connected with several genes that are homologous to bacterial cellulose synthesis (*bcs*) genes from *Acetobacter xylinus* [137]. Enzymatic digestion and chemical staining further identified this previously unknown substance in *E. coli* as cellulose. Interestingly, the regulator of the *bcs* operon is AgfD that is homologous to the curli regulator, CsgD [138]. Cellulose production is dependent on the activation of BcsA by Cyclic-di-GMP (guanosine monophosphate) [139].

#### 2.3.3. Aggregative Adherence Fimbriae (AAF)

Some *E. coli* strains form biofilm independent of curli or cellulose. Enteroaggregative *E. coli* (EAEC) can form thick aggregating biofilm on the intestinal mucosal surface and cause diarrhea [140]. EAEC strains produce aggregative adherence fimbriae (AAF) that bind to intestinal epithelial cell-matrix proteins (laminin, collagen, cytokeratin, and fibronectin). EAEC also produce 18 and 30 kDa outer membrane adhesin proteins. Upon adhesion to epithelial mucosa, EAEC secrete toxins, including enterotoxin, plasmid-encoded toxin, and enteroaggregative ST (heat stable)-like toxin, that can directly cause cell death and trigger intestinal inflammation [141]. In the absence of curli, EAEC form a unique type of biofilm (stacked-brick) in which cell-to-cell adhesion is solely mediated by AAF. AAF-mediated biofilm formation of EAEC in cell culture medium was demonstrated on two abiotic surfaces, glass and plastic [142]. In addition to their role in interaction with the MUC1 (mucus synthesis) receptor and binding EAEC to epithelial cells, AAF is also identified as a virulence factor in human infection [143]. Moreover, the binding between MUC1 and AAF also triggers upregulation of MUC1 in epithelial cells [143].

**Table 2 foods-10-02117-t002:** Bacterial factors involved in biofilm formation and pathogenesis.

Bacteria	Factors	Function		Refs
		Biofilm Formation	Pathogenicity	
*Listeria monocytogenes*	ActA (actin polymerization protein)	Bacterial sedimentation and aggregation	Rearrange host cytoskeletal structure and promote the cell-to-cell spread	[75]
	LAP (listeria adhesion protein)	Expression in recombinant *Lactobacillus* enhanced biofilm formation	Epithelial adhesion and translocation through the epithelial barrier	[50,51]
	PrfA (protein regulatory factor)	Regulate the expression of ActA that is necessary for biofilm formation	Regulatory protein that regulates the synthesis of multiple virulence factors	[71]
	WTA (wall teichoic acid)	Maintain cell wall (peptidoglycan) architecture and participate in biofilm formation	Induce inflammatory response	[77]
*Staphylococcus aureus*	Bap (biofilm-associated protein)	Adhesion to inert surfaces and intercellular adhesion in the development of biofilm formation	Establish persistent infection on a mouse infection model	[101,103]
	Protein A	Cell-to-cell adhesion in biofilm development; a major proteinaceous component in *S. aureus* biofilms	Help *S. aureus* to evade immune system in vivo	[99,100]
	PIA (polysaccharide intercellular adhesin)	Cell-to-cell binding in biofilm formation	Establish persistent in vivo infection	[91,93]
	Teichoic acid	Maintain cell wall (peptidoglycan) architecture and participate in biofilm formation	Induce inflammatory response	[76,77,111]
	FnBP (fibronectin-binding proteins)	Cell-to-cell adhesion through low-affinity homophilic interaction between neighboring cells	Promote bacterial attachment to host fibronectin for adhesion and colonization	[105,107]
	SasG (*S. aureus* surface protein G)	Zinc activated SasG-mediated biofilm formation	Adhesion to epithelial cells	[89,109]
*Salmonella enterica*	Fimbria (SEF17)	Cell-to-cell interaction in biofilm formation	Bind to human fibronectin and facilitate cell invasion	[144,145]
	Bap (biofilm-associated protein)	Bap and curli can help form strong biofilms in both biotic and abiotic surface	Colonization, intestinal persistence, invasion to liver and spleen and lethality in mice	[126]
	CsgD, BcsA	Curli and cellulose synthesis	Colonization, biofilm formation and vertical transmission to egg	[146]
*Escherichia coli*	Curli made with CsgA and CsgB	Adherence to abiotic surfaces	Adhere to epithelial cells when over expressed	[129,135]
	Fim (fimbriae)	Biofilm formation on polystyrol	Adhesion to epithelial cell lines	[136]
Enteroaggregative *E. coli* (EAEC)	Aggregative adherence fimbriae (AAF)	Mediate biofilm formation on abiotic surfaces	Bind to MUC1 on epithelial cells	[142,143]
*Pseudomonas aeruginosa*	PqsR	A key component of *Pseudomonas* quinolone signal system	Regulate the production of virulence factors, pyocyanin and hydrogen cyanide	[147]
	Flagellum	Swimming motility and biofilm formation	Flagella is an important virulence factor. The flagellum-deficient strain showed less invasion in the mouse burn wound model and less colonization in the murine intestine	[148,149]
	Type IV pili	Twitching motility, and adhesion to abiotic surfaces	Adhesion to eukaryotic cells and pathogenesis	[150]

### 2.4. Salmonella enterica Curli Fimbriae and Bap Play Important Roles in Biofilm Formation and Pathogenesis

*Salmonella enterica* is a Gram-negative rod-shaped bacterium and is closely related to *E. coli*. Based on genetic analysis, *Salmonella* diverged from *E. coli* approximately 100 million years ago [151]. *Salmonella enterica* also cause outbreaks via contaminated fresh produce, poultry, eggs, nuts, spices, flours, milk, meat and drinking water [4,36,152]. Gastroenteritis-causing *Salmonella enterica*, referred to as non-typhoidal *Salmonella* (NTS), are responsible for 1.3 billion cases and 3 million deaths worldwide [18]. In the US, 1.35 million cases, 26,500 hospitalizations, and 420 deaths occur annually [153]. Among the *Salmonella enterica* serovars, Enteritidis, Typhimurium, Newport, Heidelberg, and Javiana represent the most commonly reported outbreak causing serovars [154,155].

While the whole world was focusing on addressing the COVID-19 pandemic in 2020, an outbreak of foodborne *Salmonella* occurred. Since the beginning of July 2020, the CDC has reported 1127 cases of *Salmonella* Newport infection including 167 hospitalizations from 48 states [156]. This outbreak was unique because a very uncommon produce, red onions, was identified as the primary vehicle for the pathogen, suggesting that pathogens can adapt to new niches, making it important not to overlook the safety of any produce. Not only fresh produce can be contaminated, contamination can occur at any step along the food supply chain because bacteria can also survive for an extraordinarily long time on produce. By studying pathogen transmission from contaminated water, Kisluk et al. [157] found that *S. Typhimurium* can persist on parsley for at least four weeks when the concentration of the pathogen was above 10^8^ CFU/mL. Biofilm formation and virulence properties of *Salmonella* isolates from ready to eat shrimps showed that *S*. *Typhimurium* was the strongest biofilm former among the isolates tested [158].

Similar to *E. coli*, secreted curli from *Salmonella* forms red dry and rough (rdar) colonies on Congo red-supplemented plates [159]. *Salmonella* also form biofilms on food, the environment, and human and animal intestine [160]. Several factors are responsible for the formation of biofilm including curli, flagella, Bap, cellulose, and e-DNA [161,162,163] (Table 2). Biofilm formation by *Salmonella* has been proposed to exert anti-virulence properties [159,164]. Planktonic *S. Typhimurium* cells exhibit higher virulence gene expression than the other cells which aggregate, precipitate and show typical biofilm-related gene expression [159]. Another study revealed that the downregulation of virulence may be the result of the upregulation of Cyclic-di-GMP through the expression of *csgD* and cellulose-related genes [165].

Quorum sensing also plays an important role in *Salmonella* biofilm formation. QS molecules such as autoinducers (AI)-1, AI-2 and AI-3 are responsible for *S. Enteritidis* growth, motility, adhesion and biofilm formation [166]. The application of QS inhibitors could prevent biofilm formation and serve as an efficient therapy for controlling *Salmonella* infection [41,42].

#### 2.4.1. *Salmonella* Curli

Like *E. coli*, curli fimbriae are also a major proteinaceous extracellular fiber expressed by *Salmonella* for cell-to-cell and cell-to-surface binding [127,167]. Two unique phenotypes which helped researchers to identify a curli-expressing *Salmonella* strain are bacterial aggregation and fibronectin-binding [145]. CsgD is also the main regulator controlling the expression of the *csgBAC* operons which are responsible for the synthesis of curli fimbriae [161]. The adherent fimbriae produced by *Salmonella* Enteritidis can be classified into four types, SEF14, SEF17, SEF18, and SEF21, based on their molecular weights [145]. Specifically, the fimbrial subunits of SEF14, SEF17, SEF18 and SEF21 are expressed from genes *sefA*, *agfA*, *sefD*, and *fimA*, respectively. Austin et al. [144] showed that SEF17 is critical for stabilizing cell-to-cell interaction in biofilm formation as the SEF17-deficient mutant cannot form thick cell aggregates on the surface of either polytetrafluoroethylene or stainless steel. Fimbriae are one of the organelles that not only play a critical role in biofilm formation but also are an important factor for pathogenicity. Collinson et al. [145] found that SEF17 fimbriae but not SEF14 and SEF21 can bind to human fibronectin. Furthermore, SEF17 has been identified as a factor that significantly affects the association and invasion rate of *S.* Enteritidis on epithelial cells. Fuller et al. [168] showed that association and invasion rates of SEF17-deficient mutant were significantly reduced to approximately 13.7% and 4.2% compared to the WT strain. Antibody targeting curli has been shown to disrupt biofilm formation by *Salmonella* Typhimurium [169].

#### 2.4.2. Salmonella Biofilm-Associated Protein (Bap)

BapA is reported in *S*. Enteritidis to aid in biofilm formation [126]. Bap secretion is facilitated by the type I secretion system and the deletion of *bapA* resulted in weaker biofilm and the strain did not participate in biofilm formation when co-cultured with the parental strain. BapA expression is also coordinated with curli synthesis by *csgD* [126,161]. BapA involvement in *Salmonella* pathogenesis was shown using a *bapA* mutant strain that had reduced colonization and persistence in the intestine, invasion to liver and spleen, and lethality in mice [126]. In addition to curli fimbriae, cellulose that is the product of *bcsA*-encoded cellulose synthase is another critical component in *Salmonella* biofilms. Knockout of *csgD* and *bcsA* in a *S.* Enteritidis strain impaired biofilm formation and reduced lethality in specific pathogen-free chicken [146]. Vertical transmission from laying hen to eggs was also significantly impaired in these (Δ*csgD* and Δ*bcsA*) strains [146], suggesting a close linkage between the biofilm formation and pathogenicity in chicken.

### 2.5. Pseudomonas aeruginosa, an Opportunistic Pathogen that Forms a Strong Biofilm

*Pseudomonas aeruginosa* is a Gram-negative, non-spore-forming, and aerobic rod-shaped bacterium. It is an opportunistic pathogen affecting mostly immunocompromised individuals who are also suffering from other illnesses. Its metabolic activity is broad, and growth can occur in either nutrient-rich or nutrient-poor conditions. The ability to utilize a wide range of substrates as carbon and nitrogen sources supports the ability of *P. aeruginosa* to colonize in a variety of natural niches, such as water supply pipes and containers, and soil. Under laboratory conditions, the bacterium can grow well in a medium containing acetate and ammonium sulfate as carbon and nitrogen sources, respectively [170]. The bacterium also grew and survived in distilled water collected from a hospital, linking its involvement in nosocomial infection and spread in hospital settings [171].

*P. aeruginosa* produce a robust biofilm, which enhances its survival on different surfaces and protects the cells from other harsh conditions and treatments [148]. Another major challenge with *P. aeruginosa* is its resistance to multiple classes of antibiotics. In 2017, multidrug-resistant *P. aeruginosa* caused an estimated 32,600 infections among hospitalized patients and 2700 deaths in the US [172]. In addition, the strong biofilm-forming ability of *P. aeruginosa* makes it harder for antibiotics to access cells embedded in biofilms, which significantly reduces the effectiveness of antibiotics to treat infections.

Due to its ubiquity, persistence, and drug resistance, *P. aeruginosa* can be easily spread to humans, especially in health care settings. It can cause serious acute and chronic infections in immunocompromised people. For example, it is the most important bacterial pathogen causing progressive lung infection in cystic fibrosis patients leading to high fever, respiratory failure and death. *P. aeruginosa* can cause chronic urinary tract infections and ventilator-associated pneumonia in patients with permanent bladder catheters and intubation, respectively [173]. Treatment of *P. aeruginosa* is challenging because it can form infective biofilms after infection, which functions like a barrier protecting bacteria from complement-mediated immunity and phagocytosis, and significantly decreases the accessibility of antibiotics [174]. Therefore, comprehensive knowledge of *P. aeruginosa* biofilm formation is important for preventing and treating resistant infections.

Though *P. aeruginosa* rarely cause foodborne infection, it is still included in this review because it produces widespread biofilms in food processing plants [175] and contributes to polymicrobial biofilm formation with other foodborne pathogens posing a serious food safety concern [30]. In polymicrobial biofilms, *Pseudomonas* has been shown to support the persistence and survival of *L. monocytogenes* on conveyor belts in a salmon processing plant, which was further verified under simulated conditions [29]. These findings suggest that biofilms produced by *Pseudomonas* provide a shelter for foodborne pathogens for persistence and product contamination, and indirectly jeopardizes the safety of food products. *P. aeruginosa* is also one of the most well-studied model bacteria for bacterial biofilm research. Understanding *Pseudomonas* biofilm formation could help elucidate steps in the control and destruction of biofilms formed by other pathogens. Besides, the physiological regulation of biofilm formation and pathogenicity of *P. aeruginosa* is also similar to other bacteria. In *P. aeruginosa* biofilms, the highest cell density is arranged closest to the surface, and cells occupy only a minor fraction (approximately 2–28%) of biofilm volume, while EPS occupies the rest. Biofilm architecture is formed by EPS composed of exopolysaccharides, eDNA, and polypeptides [176]. Scanning confocal laser microscopy (SCLM) analysis has shown that *Pseudomonas* biofilms have an open and porous structure that may be designed for the transportation of nutrients and waste. Affected by the different rheological characteristics of its environment, *P. aeruginosa* can form mushroom- or pillar-like matured biofilms [177].

#### 2.5.1. Pseudomonas Flagella and Pili Aid in the Initial Attachment

The polar flagellum on *P. aeruginosa* provides swimming motility, chemotactic response, and a social movement referred to as swarming [150]. Both flagella and type IV pili are necessary for initial attachment and biofilm formation [148,178]. In the absence of pili, bacteria can form monolayers but are unable to form microcolonies, and in the absence of flagella, bacteria are defective in surface attachment [178]. In the absence of flagellar stators, MotAB or MotCD, *P. aeruginosa* produce weakened biofilm [179]. The flagellum is considered a key virulence factor because flagellum-deficient *P. aeruginosa* were less invasive in the mouse burn wound model and colonized the murine intestine less [149]. In a neonatal mouse model, *P. aeruginosa* without *fliC* caused no mortality, whereas the mortality rate of the wild-type strain was approximately 30% [180].

Various types of filamentous appendages on the bacterial surface have been studied and classified as pili [181]. Pili expressed by *P. aeruginosa* belong to the type IV pilus family, which is assembled by the polymerization of monomeric major pilin and minor pilin proteins encoded by *pilAEWV* [181]. Besides contributing to twitching motility, type IV pili are also involved in adhesion to both eukaryotic cells and abiotic surfaces [150]. Pili-mediated twitching motility is achieved by polymerization and depolymerization of pili [182,183]. *Pseudomonas* pili have a helix structure made up of a single subunit pilin of 18 kDa and each pilin monomer spontaneously assembles into filaments of 10–200 nm in length with a diameter of 1.2–5 nm [184]. The pilin genes identified in different *P. aeruginosa* strains indicate there is great heterogeneity in pilin amino acid sequences [185]. However, one common structural motif found in all *P. aeruginosa* pilin is an intrachain disulfide loop of 12–17 amino acids located at the far end of the C-terminal, which is suggested to by the site for specific binding [185]. Pili are also considered to be a major virulence factor that aids *Pseudomonas* adhesion to and invasion of epithelial cells, as demonstrated in A549 cell (lung cells) [186]. Glycosphingolipid asialo-GM1 is the host receptor for pili-mediated adhesion [187].

#### 2.5.2. Pseudomonas Microcolony Formation and Biofilm Maturation Are Regulated by a Quorum-Sensing Network

Both flagella and pili of *P. aeruginosa* are involved in the formation of cell monolayers and the typical mushroom-like structures. By expressing fluorescent proteins of different colors in the wild-type and pili-deficient *P. aeruginosa* strains, researchers visualized the distribution of these two types of bacteria and found that only the wild-type strain was located on the cap of the mushroom-like structure, suggesting that functional pili-mediated motility is necessary for forming this type of biofilm [188]. Quorum sensing (QS) systems play a critical role in the organization of cells in biofilms and the formation of rigid biofilm structures because they allow the bacterial community to globally regulate gene expression and coordinate biological processes including pathogenesis in response to population density [189]. QS is commonly applied by bacteria to direct a community’s behavior using various chemicals. This cell density-dependent cell-to-cell communication system regulates phenotypic alterations at the early stages of biofilm formation after attachment [190]. Currently, four types of quorum sensing systems that regulate the expression of biofilm formation have been identified in *P. aeruginosa*: the Las, Rhl, PQS, and IQS systems [191]. Each system contains at least two major functional elements, one category senses the critical concentration of a specific autoinducer (AI), and serves as a transcriptional activator for genes encoding the second category—cognate AI synthases [192]. The Las and RhI systems are triggered by an increase in cell density during the early exponential growth phase, while PQS and IQS systems are activated during late exponential growth [193].

The Las system involves the production of an autoinducer N-(3-oxododecanoyl)-L-homoserine lactone (3-O-C_12_-HSL), which is regulated by the AI synthases LasI, and sensed by the transcription factor LasR. The Rhl system uses AI synthase RhlI to produce an autoinducer, N-butanoyl-L-homoserine lactone (C_4_-HSL), which uses RhlR as its cognate receptor. Both Las and RhI are essential regulating systems for the maturation of *Pseudomonas* biofilm and their inhibition using synthetic molecules prevented pathogenesis in animal models [43].

To explore the relationship between quorum-sensing systems and biosynthesis of biofilm matrix, another research team focused on the expression of *pel* and the quorum sensing system *las* [194]. The *pel* cluster consists of seven genes that are responsible for biosynthesis of polysaccharides, a major component making up the extracellular matrix of biofilms [195,196,197]. A second genetic locus, *psl*, was identified to be responsible for producing mannose-rich extracellular materials, thus it was found that *P. aeruginosa* requires at least one of these loci, *pel* or *psl*, to form normal biofilms [196]. Gene expression of *pel* is regulated by the *las* quorum-sensing system [194] and induces the expression of *pel* genes to produce an extracellular matrix. Furthermore, the *rhl* quorum-sensing system may be involved to a lesser extent in the induction of *pel* genes. Cationic exopolysaccharide Pel binding to negatively charged eDNA plays an essential role in maintaining the integrity of biofilms [198,199].

The *Pseudomonas* quinolone signal (PQS) system is also involved in the release of eDNA and biosurfactants, which are essential for the development of mature biofilms during late exponential phase [200]. The *pqsA* mutant strain contains less eDNA than biofilms formed by their wild-type counterpart [201]. PqsR is also necessary for the synthesis of PQS as a positive regulator of *pqsA* expression by binding to the *pqsA* promoter [147]. The binding of PqsR to the promoter is further increased in the presence of PQS, suggesting that PQS may be a cofactor for PqsR. PqsR also controls the synthesis of more than 60 types of secreted anthranilic acid derivatives [202]. Production of virulence factors, including pyocyanin and hydrogen cyanide, also requires PqsR, which gives PqsR to play a key role in both pathogenicity and biofilm formation [189].

The PQS system is important in virulence factors generation during biofilm development. PQS mutants showed reduced biofilm development and less production of virulence factors, such as pyocyanin, elastase, lectin, and rhamnolipids. The correlation between the PQS system and infectivity has been tested using several in vivo models. In burn-wound mouse models, the survival rate of mice infected with *pqsA* mutant strains was approximately 50% higher than infection with the parental strain [203]. *P. aeruginosa* is also an opportunistic plant pathogen. A mutant strain with dysfunctional PQS production grew dramatically less than the wild-type strain in *Arabidopsis*, suggesting a critical role of PQS for the overall pathogenicity of *P. aeruginosa* [204].

The integrated quorum sensing (IQS) system is less studied compared to the other three systems. It produces 2-(2-hydroxyphenyl)-thiazole-4-carbaldehydte (aeruginaldehyde) as its cognate AI, while the receptor has not been found. A non-ribosomal peptide synthase gene cluster *ambBCDE* is responsible for IQS synthesis. This disruption led to a decrease in PQS and BHL production, along with other virulence factors such as pyocyanin, rhamnolipids, and elastase [205]. However, Cornelis [206] recently commented that *amb* gene cluster is not responsible for aeruginaldehyde production, since its production is also found in *Pseudomonas* strains lacking this cluster. The important contribution of IQS to virulence of *P. aeruginosa* has been shown in animal models. For instance, mice infected by *P. aeruginosa* without *ambB* and *lasI* genes had a higher survival rate than either the wild-type or a *lasI* knockout mutant strain [207]. IQS production is related to phosphate availability in the host, suggesting that the IQS system may be responsible for adjusting virulence during infection [208]. Under iron- and phosphate-deficient conditions, both PQS and IQS systems could be enhanced, which will lead to increased virulence factor synthesis, causing increased mortality of the host organism.

In addition, in mixed-culture biofilms of *P. aeruginosa* and *S. aureus*, the presence of the latter organism can also increase exotoxin A expression [209,210], indicating that expression of virulence genes by one species in biofilms can be altered by the presence of another species.

## 3. Conclusions and Future Perspectives

In summary, multifunctional molecules involved in both bacterial pathogenesis and biofilm formation demonstrate a close connection between the two aspects. In *L. monocytogenes*, ActA rearranges actin in the host cell cytosol to propel cell-to-cell movement and also initiates biofilm formation by precipitating bacteria. Likewise, teichoic acids responsible for maintaining Gram-positive bacterial cell architecture also induce inflammatory response during infection and contribute to biofilm formation in both *L. monocytogenes* and *S. aureus*. Protein A of *S. aureus* not only helps the pathogen to evade the immune system but also facilitates cell-to-cell adhesion in biofilm development. Other proteins, including FnBP, SasG, and Bap, are also responsible for biofilm formation and pathogenesis in *S. aureus*. Curli is critical for biofilm formation and pathogenesis in *E. coli*. Similarly, curli and Bap are important in biofilm formation, intestinal colonization, and pathogenesis in gastroenteritis-causing non-typhoidal *Salmonella*. In *Pseudomonas*, PqsR plays a key role in the *Pseudomonas* quinolone signaling system and also regulates the production of virulence factors promoting bacterial biofilm formation and attachment to host epithelium. Other factors including flagella and type IV fimbriae are important in biofilm formation and colonization on epithelial cells. Many of the virulence factors that are involved in biofilm formation and host cell colonization have redundant functions, suggesting that even in the absence of one factor, bacteria can still form biofilms that are food safety and public health concerns.

Although the pathogenesis of multiple foodborne pathogens has been comprehensively studied, most of the results were generated using planktonic cultures under laboratory conditions. The actual risk of consuming pathogens from biofilms should be further characterized using animal models instead of only in vitro cultured mammalian cell models or virulence factor expression analyses. Recently, we used *L. monocytogenes* as a model foodborne pathogen to investigate the virulence of the bacteria in biofilms. Our data indicate that the virulence of biofilm-isolated *L. monocytogenes* was upregulated after 48 h bacterial adaption to the intestinal environment. These findings enhanced our understanding of bacterial pathogenesis of biofilm-isolated bacteria, and these data should be beneficial for the accurate evaluation of biofilm risks in food processing environments. Similarly, the assessment of the pathogenicity of other foodborne pathogens, such as *E. coli* and *Salmonella*, isolated from biofilms could also be further investigated using animal models. Using bacteria isolated from biofilms could also be a good model for studying bacteria switching from a saprophytic lifestyle to pathogenic status in animal hosts.

Although there are many studies of biofilm formation on plastic, stainless steel, or glass surfaces, more in-depth studies are needed of foodborne pathogen biofilms formed directly on food surfaces, for example, cantaloupe skin or eggshell. Bacteria isolated from these biofilms should represent a more realistic model to assess the risk of consuming foodborne pathogens found on food surfaces.

## Figures and Tables

**Figure 2 foods-10-02117-f002:**
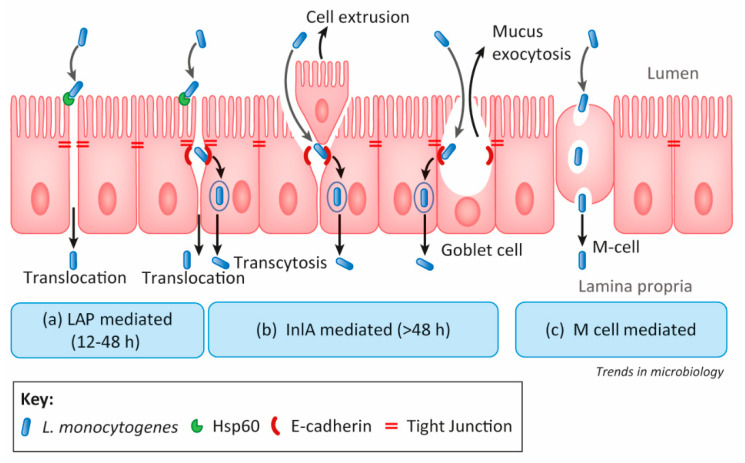
*Listeria monocytogenes* translocation pathways from lumen to lamina propria in the intestine. Figure adapted with permission from Drolia and Bhunia 2019 [47]. Copyright, Elsevier.

**Figure 3 foods-10-02117-f003:**
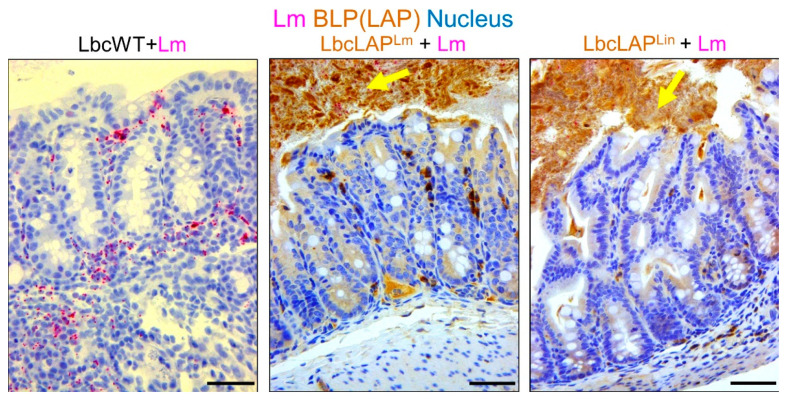
Biofilms formed (above) by recombinant *Lactobacillus casei* (Lbc) expressing *Listeria* Adhesion Protein (LAP) from *L. monocytogenes* (LbcLAP^Lm^) or nonpathogenic *L. innocua* (LbcLAP^Lin^) on mouse colonic villi after feeding for ten days (arrows). The wild-type *Lactobacillus casei* (LbcWT) did not show any biofilm formation (left panel). Bar, 25 µm. The figure was adapted with permission from Drolia et al. 2020 [51].

**Figure 4 foods-10-02117-f004:**
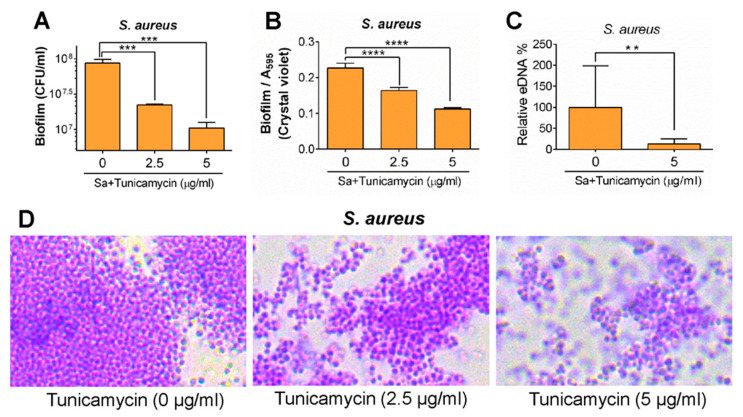
Tunicamycin-mediated inhibition of teichoic acids also suppresses *S. aureus* biofilm formation in a dose-dependent manner. Upper panels (**A**–**C**) show a quantitative assessment of biofilm formation (colony counts, crystal violet staining, and eDNA amounts), and lower panel (**D**) shows crystal violet-stained *S. aureus* cells in biofilms in the presence of variable concentrations of tunicamycin. The figure was adapted with permission from Zhu et al. 2018 [77]. ****, *p* < 0.0001; ***, *p* < 0.001; **, *p* < 0.01.

**Table 1 foods-10-02117-t001:** Summary of foodborne pathogens and their implication in food safety.

Pathogen	Gram Stain	Spore Forming	Foods Involved	Infectious Dose	Disease Symptoms
*Listeria monocytogenes*	Positive	No	Ready-to-eat meat, dairy, fish, fruits, and vegetables. Foods with high protein content, such as deli meat, fish, and cheese.	<100–10^11^ CFU (colony forming unit) depending on individual immunological health [31,32].	To healthy children and adults, flu-like symptoms include diarrhea, fever, vomiting, joint pain, headache. Invasive systemic disease in the immunocompromised host. Miscarriage and stillbirth in pregnant women. Meningitis or encephalitis in newborns and elderly.
*Staphylococcus aureus*	Positive	No	Milk products, meat, and hand-prepared foods.	*S. aureus* cells: 10^5^–10^8^ CFU/g. Toxin: 1 ng/g [33].	Vomiting, diarrhea, and sometimes toxic shock symptoms including fever, low blood pressure, and even death.
*Escherichia coli*	Negative	No	Meat products, such as ground beef and sausage, fruits and vegetables.	Cause both food poisoning and infection. As low as 50–100 CFU of Enterohemorrhagic *E. coli* (EHEC) can cause infection [34].	Vomiting, diarrhea, bloody diarrhea, hemorrhagic colitis, hemolytic uremic syndrome.
*Salmonella enterica*	Negative	No	Poultry products, meat, fish, vegetables, nuts, flours, milk, and drinking water.	Approximately 10^3^–10^5^ CFU is needed to cause diseases [35]. However, as low as 1–100 CFU is also implicated depending on the serovars involved [36].	Typhoid fever, fever, vomiting, diarrhea, abdominal pain. It causes invasive disease in immunocompromised patients.
*Pseudomonas aeruginosa*	Negative	No	Not a common foodborne pathogen but may present in water, soil, plants, and foods. It contributes to the polymicrobial biofilm formation with other foodborne pathogens to be a food safety concern.	An opportunistic pathogen and infectious dose are highly variable; 10^3^–10^9^ CFU [37].	Cause serious diseases in burn and cystic fibrosis patients with fever, chills, coughs with yellow, green, or bloody discharge. Gastroenteritis and diarrhea in some patients.

## Data Availability

Not applicable.

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
