# Peer review of "Bacterial Biofilms and Their Implications in Pathogenesis and Food Safety"

_foods, 2021, doi:10.3390/foods10092117_

Round 1

Reviewer 1 Report

The manuscript is interesting and provides important information on the formation of biofilm by foodborne bacteria. The manuscript was prepared on the basis of 205 references, 26% of which were from the last 5 years and 53% from the last 10 years. The remaining articles (about half) are older than 10 years. Authors should prepare the manuscript on the basis of more recent research than 10 years, especially based on original research.

Lines 30 - 101, Introduction: Has been compiled from 24 articles, of which only 4 are from the last 5 years. Information should be from recent years and the introduction should be corrected, eg lines 72-73: The number of deaths has been given 11 years ago [14, 15]. Please provide newer data, maximum 5 years ago.

Reviewer 2 Report

Dear Authors

Please note that english is not my native language.

Thank you for your good revision about bacterial biofilms, one of the chalenges to food safety. However, there are some aspects that need be improved.

The first one is the revision by an english native speaker.

MAJOR COMMENTS

In my opinion, Pseudomonas aeruginosa has no place in this paper. You included it  “as a model used for Gram-negative bacterial biofilm research” (line 100). I don't think it's an appropriate justification because you describe other Gram-negatives (Escherichia coli e Salmonella enterica). Besides Pseudomonas aeruginosa is not considered as a food borne pathogen.  Pseudomonas aeruginosa is environmental, ubiquitous and capable of forming biofilms, but the different diseases it causes are not due to food intake.

Table 1 -  Foods involved in Listeria monocytogenes – “Foods with high protein content, like ice cream and cheese.”

Cheese has a high protein content but ice cream does not.  Ice cream has a high content in fat and carbohydrates. As is mentioned in the published literature, Listeria monocytogenes is found in ice cream because, although it does not multiply, it remains viable for long periods in freezing temperatures. When the temperature rises, it recovers its activity. Please, confirm.

Table 1 – Food involved in Escherichia coli and in Salmonella enterica  

The drinking water is an important vehicle of these microrganisms. Why didn’t you include it?

Table 1Salmonella enterica – Infectious dose.

Infectious dose of Salmonella depends on the serotype. It may be only 1cfu/g or mL, 10 – 100 cfu/g or mL …

Please see, van den Brom, 2020 -  https://doi.org/10.1016/j.smallrumres.2020.106123

MINOR COMMENTS

Lines 144-145 – “Listeria monocytogenes epitelial barrier…” This sentence may be useful but I don't think it's well located here.

Line 196 – “2.1.2 – Role of Sigma…”  It may be 2.1.1 – Role of Sigma…” Before line 196 I did not find 2.1.1.

Line 198 – You mentioned Table 2 but there are two Table 1. Please correct line 525 – “Table 1 – Bacterial factors…” to Table 2 – Bacterial factors …

Line 318 – “…synthesized byproducts…” I think you can say “…synthesized by products

Reviewer 3 Report

Manuscript very well and comprehensively written. It addresses important issues regarding biofilms formed by major pathogens. Thematically, it is most appropriate for publication in Foods. The manuscript needs only minor corrections which I have included in the attached pdf file. Once these have been addressed, it can be considered for publication in Foods with the full confidence of the reviewer.
